# Model-Based Dynamic Toll Pricing: An Overview

Claudio Lombardi [1,2,*] , Luís Picado-Santos [1,*] and Anuradha M. Annaswamy [2,*]

1. CERIS, Instituto Superior Técnico, Universidade de Lisboa, Av. Rovisco Pais 1, 1049-001 Lisboa, Portugal
2. Active Adaptive Control Laboratory, Department of Mechanical Engineering, Massachusetts Institute of Technology, 77 Massachusetts Avenue, Cambridge, MA 02139, USA
* Correspondence: claudio.lombardi@tecnico.ulisboa.pt (C.L.); luispicadosantos@tecnico.ulisboa.pt (L.P.-S.); aanna@mit.edu (A.M.A.)

**Abstract:** In this paper, we review some of the most recent research regarding design, simulation, implementation and evaluation of dynamic tolling schemes. Analyzing the structure of the reviewed studies, we identify the common elements and the differences in the approaches chosen by different authors, presenting an overview of the methods for price definition and of the simulation techniques as well as a discussion on the newest technology applications in the field. Optimization revealed to be the dominant price definition method, while control-based algorithms are notably employed for managed lanes toll pricing schemes. Regarding traffic and driver behavior simulation we observed a great variety of solutions throughout the reviewed papers, with a prevalence of macroscopic models for the former and logit models for the latter. Still few papers include models for externalities quantification, while AI paradigms are gaining importance in the field.

**Keywords:** congestion management; dynamic toll pricing; control-based algorithms; optimization; driver behavior model; traffic model; externalities quantification

## 1. Introduction

During the last decades, the technique of dynamic pricing has started to play a key-role in manifold domains of Intelligent Transportation Systems, such as fare pricing, charging/discharging pricing for electric vehicles, parking pricing and congestion pricing [1]. This technique entails the variation of prices according to market conditions, in response to the demand-supply imbalance. Dynamic toll pricing is a variant of congestion pricing [1] where tolls vary in real-time as a function of current traffic conditions, as opposed to flat tolls, which stay constant over time, and scheduled tolls, where tolls vary by time of day, day of the week or season following a predetermined schedule [2].

The idea of traffic congestion pricing was introduced in the 1920s by the pioneering work of Pigou [3], but its first implementation was only in 1975 with the Singapore Area Licensing Scheme [4]. Congestion pricing can be applied at different scales, from single lanes to large scale networks.

In Figure 1, we can see the distribution by scope of the dynamic toll pricing studies analyzed in this review, revealing the predominance of studies concerning managed lanes, followed by networks and, eventually, general tolled facilities.

Managed lanes, i.e., High Occupancy Toll (HOT) and Express lanes facilities constitute one of the few examples of congestion toll pricing operating schemes; as of January 2019, dynamic toll pricing is implemented on at least 13 managed lanes facilities in the United States [5] and their schemes are considered to be the closest to pure dynamic toll [6], while, at the moment, there are no implemented dynamic city tolling systems [7]. Although most of the adopted dynamic toll pricing strategies are simple and heuristic [8], some of the existing schemes showed to have better performances than fixed tolls [9,10], encouraging researchers to develop new schemes.

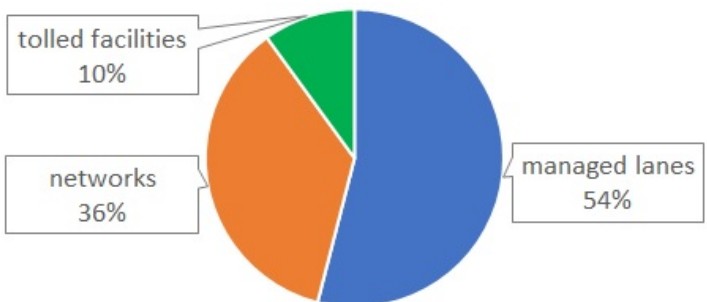

**Figure 1.** Distribution of the analyzed dynamic toll pricing studies by scope.

To present a brief overview of the research efforts carried out worldwide on dynamic toll pricing, we evaluated the studies published to date. This analysis was based on the Scopus search engine (www.scopus.com (accessed on 3 May 2021)). The query string was written to find documents matching the keywords 'dynamic' AND 'toll' AND 'pricing' in the title, the abstract or the keywords. This query provided 209 references in all, spanning from 1992 to 2021. For comparison, we show the results of other query runs (Table 1), using different combinations in pairs of the same keywords.

**Table 1.** Results of three different runs of the keywords search in title, abstract or keywords of documents on the Scopus search engine, characterized by different keywords combinations.

| Keywords | References | Years (Since) |
|---|---|---|
| dynamic AND toll AND pricing | 209 | 1992 |
| dynamic AND toll | 1503 | 1973 |
| dynamic AND pricing | 8894 | 1967 |

The analysis of the outcomes of the query indicates that dynamic toll pricing embraces a very diverse group of subject areas with a prevalence of engineering, social science, computer science and mathematics, as appears in Table 2. A classification of the documents by year, in Figure 2, suggests an overall increase of the research efforts on dynamic toll pricing during the last decades, with a peak of published documents in 2018. In this review, we will focus in particular on the research efforts carried out since 2008 when existing research on pricing algorithms for HOT operations was considered to be in its early stage [11]. Sorting of the documents by country (Figure 3) revealed that the USA and China are the countries that have contributed the most to the number of documents published on dynamic toll pricing.

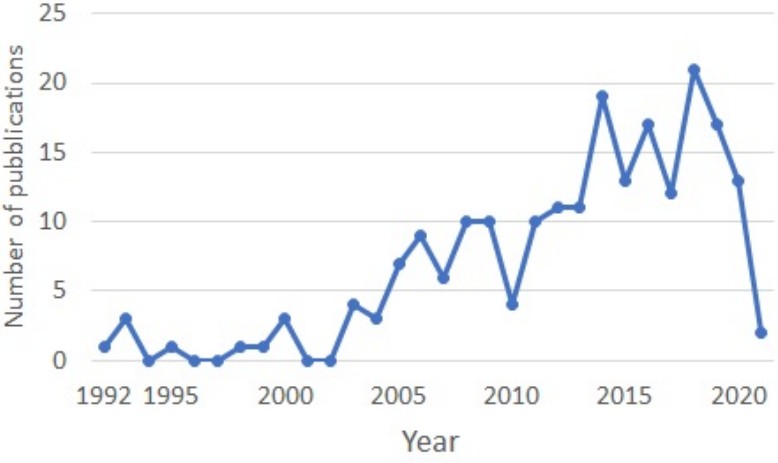

**Figure 2.** Evolution of the number of publications on dynamic toll pricing.

**Table 2.** Analysis of the search results in terms of subject areas, from Scopus.

| Subject Area | Documents |
|---|---|
| Engineering | 151 |
| Social Sciences | 111 |
| Computer Science | 72 |
| Mathematics | 21 |
| Environmental Science | 12 |
| Economics, Econometric and Finance | 11 |
| Decision Sciences | 9 |
| Business, Management and Accounting | 6 |
| Physics and Astronomy | 5 |
| Energy | 4 |
| Earth and Planetary Science | 3 |
| Arts and Humanities | 1 |
| Chemical Engineering | 1 |
| Materials Science | 1 |

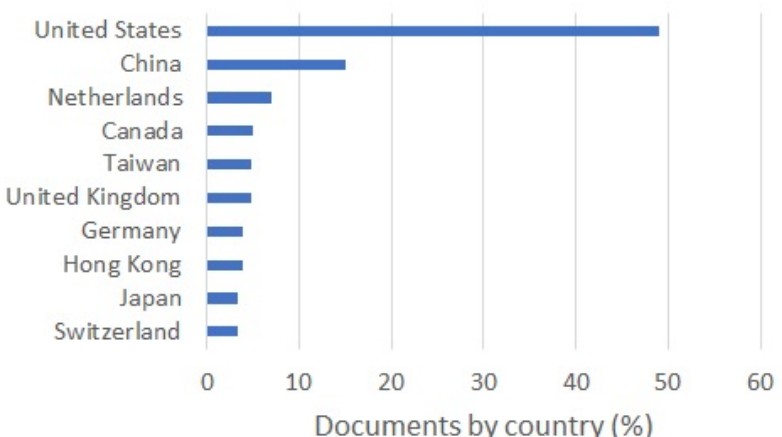

**Figure 3.** Distribution of the published documents on dynamic toll pricing by country.

In almost all approaches evaluated in this survey, the overall structure consists of a pricing strategy followed by a simulation study that evaluates the pricing strategy. This simulation study is carried out using a socio-technical model that combines a driver behavior model and a traffic flow model [12], with some papers also including quantification of traffic externalities.

A list of the main reviewed dynamic toll pricing studies described by reference, year of publication, scope, main pricing rule principle, traffic simulation basis, driver behavior model and recent technology application is presented in Table 3. All the quantitative considerations about the popularity of different scopes and methods presented in this paper are determined based on the papers mentioned in Table 3.

In Section 2, we introduce an overview of the most relevant methods for price definition, in Section 3, we review some of the adopted simulation techniques, in Section 4, we carry out a discussion about the interactions of the newest technology applications with dynamic toll pricing, and in Section 5 we present our conclusions with some insights about future research perspectives.

**Table 3.** A list of the main reviewed studies described by reference (Ref.), year of publication, scope, main pricing rule principle, traffic simulation basis, driver behavior model and recent technology application. We included in the table also some studies which do not give a complete pricing scheme, but are limited to the presentation of some tool which can be combined with other models in order to obtain a complete pricing scheme. The following abbreviations are adopted in this table: AD = Arrivals-Departures, adj. = adjustable, av. = average, bottl. = bottleneck, CTM = Cell Transmission Model, D2D = Day-to-day, discr. = discrete, distr. = distribution, DTC = departure time choice, DUE = Dynamic User Equilibrium, emb. = embedded, enh. = enhanced, fac. = facilities, Gauss. = Gaussian, infl. = inflow, log-n. = log-normal, LS = large scale, LWR = Lighthill-Witham-Richards, macro = macroscopic, max. = maximizing, MC = multiclass, meso = mesoscopic, MFD = macroscopic fundamental diagram, micro = microscopic, min. = minimizing, MM = multimodal, MN = multinomial, na. = national, NW = networks, obj. = objective, op. = optimal, oper. = operator, opt. = optmization, P = proportional, PD = porportional derivative, PI = Proportional Integral, PQ = Point-Queue concept, PWcontr. = piece-wise linear control, PID = Proportional Integral Derivative, rd = road, RF = random forest, RL = reinforcement learning, sens. = sensitivity, SRDC = simultaneous route-and-departure choice, syst. = system, TDGP = total delay on general purpose lanes, TE = total emissions, throughp. = throughput, transp. = transportation, TTC = total travel cost, TTT = total travel time, TTTC = total travel time cost, UE = User Equilibrium, unt. = untolled, urb. = urban, vehic. = vehicular, VOT = value of time, Weib. = Weibull.

| Ref. | Year | Scope | Pricing Rule | Traffic Simulation | Driver Behavior | Recent Technology |
|------|------|-------|--------------|--------------------|-----------------|-------------------|
| [13] | 2009 | managed lanes | P control | delay operator (PQ) | binary logit | self-learning |
| [11] | 2008 | HOT lanes | PWcontr. | micro (VISSIM) | binary logit | - |
| [14] | 2018 | managed lanes | PWcontr. op. gains | micro (VISSIM) | agent-based | - |
| [15] | 2014 | managed lanes | PWcontr. op. gains | micro (VISSIM) | agent-based | - |
| [16] | 2015 | managed lanes | PWcontr. op. gains | micro (VISSIM) | binary logit | - |
| [17] | 2016 | HOT lanes | PD control | macro (LWR-based) | binary logit | - |
| [18] | 2016 | MM urb. NW | PI control | macro (MFD) | agent-based | - |
| [19] | 2018 | LS NW | PI control | macro (MFD) | C-logit | - |
| [20] | 2014 | HOT lanes | PI control | macro (LWR) | binary logit | - |
| [21] | 2015 | HOT lanes | PID control | micro (Paramics) | agent-based | - |
| [22] | 2018 | HOT lanes | PD control | macro (LWR-based) | binary logit | - |
| [23] | 2016 | HOT lanes | cascaded control | micro (VISSIM) | VOT distr. (Gauss.) | - |
| [24] | 2017 | toll lanes | optimal control | queuing theory | binary logit | - |
| [25] | 2018 | toll lanes | optimal control | queuing theory | binary logit | - |
| [26] | 2018 | LS system | optimal control | discretized model | UE | - |
| [27] | 2008 | freeways | optimal control | micro (Paramics) | emb. in Paramics | - |
| [28] | 2013 | toll facilities | revenue-max. opt. | linear function | binary logit | - |
| [29] | 2015 | tolled route | revenue-max. opt. | micro (CORSIM) | VOT distr. (Weib.) | - |
| [30] | 2019 | managed lanes | revenue-max. opt. | micro (MITSIM) meso (DynaMIT) | path-size logit | - |
| [31] | 2011 | HOT lanes | throughp.-max. opt. | macro (CTM [32]) | binary logit | self-learning |
| [8] | 2013 | HOT lanes | av.flow-max. opt. | macro (CTM [32]) | binary logit | self-learning |
| [33] | 2015 | toll. NW (D2D) | bi-objective opt. | UE | UE | - |
| [34] | 2014 | transp. NW | syst.cost-min. | delay operator | VOT distr. | - |
| [35] | 2009 | general NW | bi-level opt. | UE | UE | - |
| [36] | 2018 | two-layer NW | bi-level opt. | macro (MFD) | UE micro (VISSIM) | - |
| [37] | 2019 | NW | bi-level opt. | DUE | DUE | - |
| [38] | 2012 | NW | bi-level opt. | DUE | DUE | - |
| [39] | 2016 | NW (D2D) | Markov DP | Markovian | path-size logit | - |
| [40] | 2018 | managed lanes | Markov DP | macro (CTM) | VOT distr. (disc.) | - |
| [41] | 2018 | managed lanes | Markov DP | macro (CTM) | agent-based | MA RL |
| [42] | 2020 | express lanes | Markov DP | macro (CTM) | MC binary logit MC decision route | deep RL |
| [43] | 2013 | LS MM NW | game theory | delay oper. (ABM) | Nash Equilibrium | - |
| [44] | 2014 | HOT lanes | game theory | delay oper. (AD) | VOT distr. (log-n.) | - |
| [45] | 2014 | HOT lanes | multi-obj. opt. | delay oper. (AD) | VOT distr. (log-n.) | - |
| [46] | 2012 | managed lanes | revenue-max. opt. | macro [47] | binary logit | - |
| [48] | 2015 | HOT lanes | opt. (any objective) | macro (CTM) | binary logit | - |
| [49] | 2020 | HOT lanes | throughp.-max. opt. | delay operator (PQ) | MN logit UE VOT distr. general lane-choice | - |

**Table 3.** *Cont.*

| Ref. | Year | Scope | Pricing Rule | Traffic Simulation | Driver Behavior | Recent Technology |
|---|---|---|---|---|---|---|
| [50] | 2012 | urb. NW | P control | macro (MFD) | binary logit | - |
|  |  |  |  |  | agent-based | - |
| [51] | 2017 | large urb. NW | outflow-max. opt. | macro (MFD) | UE | - |
| [52] | 2013 | HOT lanes | P control | micro (CORSIM) | emb. in CORSIM | - |
| [53] | 2017 | toll roads | multi-obj. opt. | meso (DynusT) | - | - |
| [54] | 2016 | large NW | bottl. model [55] | meso | econometric DTC | - |
| [56] | 2017 | large NW | TTT-min. opt. | meso | econometric DTC | - |
| [57] | 2015 | toll vs. unt. fac. | various obj. opt. | delay operator (AD) | UE | - |
| [58] | 2015 | HOT lanes | TTTC-min. opt. | delay opertor (BPR) | time and price sens. | - |
|  |  |  | TTT-min. opt. |  |  |  |
| [59] | 2017 | urb./na. rd NW | shifting obj. | - | - | big data mining |
| [6] | 2019 | urb./na. rd NW | shifting obj. (adj.) | delay opertor (BPR) | binary logit | - |
| [12] | 2013 | HOT lanes | HOT infl.-max. opt. | delay operator (AD) | VOT distr. (Burr) | - |
| [60] | 2013 | toll. vehic. NW | TTC&TE-min. opt. | SRDC (DUE)[61] | SRDC (DUE)[61] | - |
| [62] | 2019 | HOT lanes | TDGP-min. opt. | delay operator (AD) | RF predictions | RF predictions |

## 2. Overview of Dynamic Toll Price Definition Methods

The dynamic tolling operations schemes currently in force in the managed lanes facilities in the United States are proprietary, and therefore not available to the public [63]. A notable exception is the algorithm employed to compute MnPASS dynamic toll is published [64]; MnPASS was one of the first managed lanes facilities to become dynamically tolled in 2005 [65] creating a congestion-free lane for both single occupancy vehicles whose drivers are willing to pay the toll and high occupancy vehicles [66]. During the morning and evening peak hours, the toll rate for a particular entry point of the MnPASS express lanes is a function of the maximum density downstream of the entry point: variations of this density define a specific variation of the toll based on lookup tables [64] with a 3-min tolling interval [13].

Two operational objectives are identified for HOT lanes in [49]: (i) to keep free-flow conditions to ensure travel time reliability; and (ii) to maximize the tolled lanes' throughput in order to minimize the system's total delay. The most relevant methods for dynamic toll price definition explored in literature that we present in the remainder of this section are usually built to pursue one of these two objectives or a combination of both. We can say that control-based approaches are more focused on the former objective, while optimization algorithms usually pursue the latter or the maximization of another performance index of the tolled infrastructure. Figure 4 displays the distribution of the analyzed studies categorized by scope and main price definition method. We note that optimization (OPT) is the predominant price definition method, especially for studies regarding networks (NW) and tolled facilities (TF). For managed lanes studies (ML), which include also research concerning HOT lanes, express lanes and general toll lanes, control-based algorithms (C) are adopted in almost as many cases as optimization-based ones. We also stress that optimal control, which may be considered an application of both optimization and control theories, is implemented in two papers relating to managed lanes, one relating to network and one relating to toll facilities and we counted these algorithms in the control-based groups. Three approaches that are not attributable either to optimization or control theory, identified with label OTH, are employed for networks.

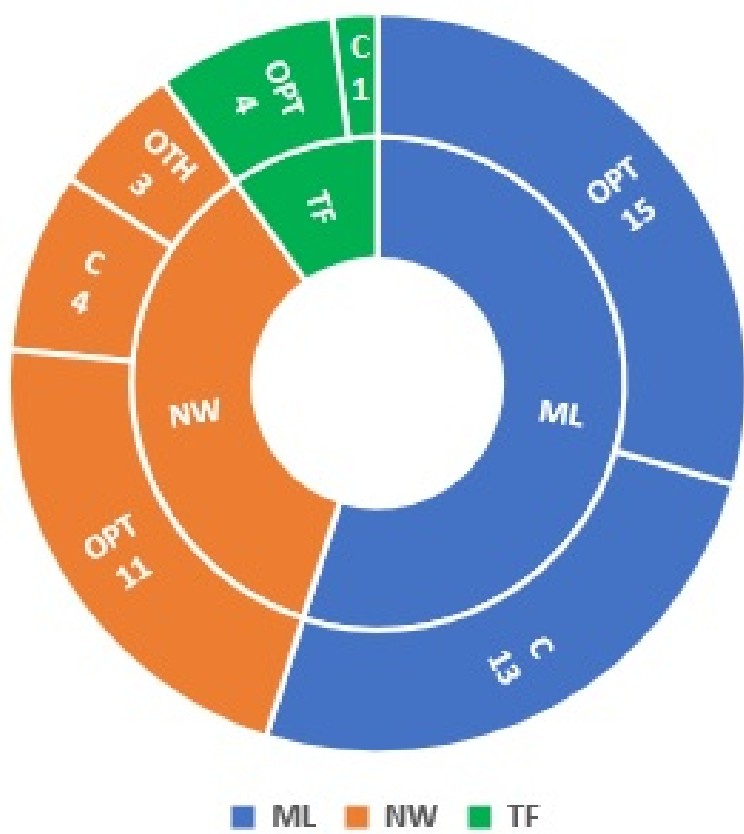

**Figure 4.** The distribution of the analyzed studies categorized by scope (inner circle) and main price definition method (outer circle). Labels: ML = Managed Lanes, NW = Networks, TF = Tolled Facilities, OPT = Optimization, C = Control, OTH = Other.

In what follows, Section 2.1 is dedicated to control-based algorithms and Section 2.2 to optimization-based approaches.

### 2.1. Control-Based Algorithms

The general topic of traffic modeling and control has been studied at length in the literature (ex. [67]). Broadly speaking, traffic control can be classified as either road-based or vehicle-based. Road-based traffic control includes ramp management via methods such as ramp metering [68,69], posting of variable speed limits so as to homogenize speeds and avoid stop-and-go waves [70,71], average speed enforcement [72,73], lane control redirecting vehicles to different lanes [74], Route Guidance and Information Systems (RGIS) communicated to drivers through various messaging methods [75,76]. A fairly large literature exists on vehicle-based control, which pertain to Vehicle-to-Infrastructure (V2I) based communication strategies and Connected and Automated Vehicles (CAVs) based coordination strategies [77]. In addition to the above methods, one can consider a dynamic toll-pricing based approach, which may be viewed as either road-based or vehicle-based, where the toll price is messaged to the driver (either through a message posted on the road or directly to the vehicle), and the driver makes a decision in turn whether to enter the highway or not [17,22]. Viewing real-time traffic information data as a sensor and the toll based on these data as a control input, the underlying traffic congestion problem is posed as a control problem, where the driver can be viewed as an actuator. The concept of transactive control, a feedback control strategy enabled through economic contracts to incentivize and enable flexible consumption, is eliciting significant attention of late [78], with multiple applications in dynamic toll pricing.

A simple feedback digital control law for on-ramp metering called ALINEA was proposed in [79] as in (1):

$$r(k) = r(k-1) + k_p \cdot [o_{ref} - o(k)]. \tag{1}$$

The metering rate at time $k$, $r(k)$ (in equivalent passenger vehicles/hour, or veh/h), is defined by the product of a proportional gain $k_p$ with the difference between the desired value of an occupancy parameter $o_{ref}$ and its current value $o(k)$ (in % km/km), which, in turn, is related to traffic density $\rho(k)$ (in equivalent passenger vehicles/km, or veh/km) by Equation (2):

$$\rho(k) = \eta \cdot o(k), \tag{2}$$

where $\eta$ is defined as in (3):

$$\eta = \frac{\nu}{100 \cdot l}, \tag{3}$$

with $\nu$ equal to the number of lanes on the mainstream and $l$ (in km/veh) equal to the mean effective vehicles length in km.

A principle similar to ALINEA was applied in [13] to build a toll pricing controller, given by Equation (4):

$$\pi(k+1) = \pi(k) + k_p \cdot [o(k) - o_{ref}], \tag{4}$$

where the toll price $\pi(k)$ (in monetary units) is defined by a controller, being the control variable an occupancy parameter. A similar concept was applied in [11], where the optimal HOT flow ratio to set the dynamic toll is defined by a control-based piecewise linear function, in [50] and in [52]. Also in [14–16] the control mechanism is based on a piecewise linear function but with variable revenue-maximizing optimal gains.

In [17], desired flow into the HOT at time $t$ (in veh/h), indicated with $y(t)$, is computed through a proportional derivative (PD) analog controller with a feedforward additive component (5):

$$y(t) = k_p \cdot e(t) + k_d \cdot \dot{e}(t) + k_{ff} \cdot \rho_{ref}, \tag{5}$$

where $k_p$, $k_d$ and $k_{ff}$ represent proportional, derivative and feedforward gains, respectively, and $e(t)$ is an error signal, given by the difference between the desired density $\rho_{ref}$ and the current density $\rho(t)$ (in veh/km).

In [18], in turns, a proportional integral (PI) controller defines time-dependent tolls for road networks based on the congestion level of the network (proportional part) and on users' adaption to the tolls (integral part). A PI feedback control for toll pricing extendable to time-dependent pricing for a large scale dynamic traffic network is presented in [19] and a PI adaptive controller computes desired HOT flow rates in [20].

The HOT dynamic pricing algorithm proposed in [21] embeds a proportional integral derivative (PID) controller that considers the time delay between the tollbooth and the exit of HOT lanes.

In [80] we propose a dynamic toll pricing scheme based on feedback control theory, similar to the approach in [17,22], for a freeway segment with multiple access locations.

Cascaded feedback control is employed to build a discrete-time dynamic tolling computation algorithm that later was successfully implemented in a HOT between Ben Gurion Airport and Tel Aviv, in Israel [23]. Without going into the details of the equations, the inner loop is a PD controller which computes the toll rate increment having the inflow to the HOT as feedback signal, while the outer loop is given by a PI controller that calculates the inflow reference value with the average velocity on the HOT lane as the feedback signal.

In optimal control, the control gains are optimized so that some performance measure of the system, function of its current and past states and inputs, is maximized. Hamilton-Jacobi-Bellman equation is used to design feedback optimal control laws that utilize real-time traffic information for defining dynamic tolls of lanes or routes in [24–26]. The extended Kalman filtering technology is employed in [27] for a stochastic optimal control

based algorithm to execute a freeway traffic management mechanism that integrates dynamic toll and ramp control.

## 2.2. Optimization-Based Algorithms

While control-based dynamic toll pricing approaches described in Section 2.1 usually focus on keeping free-flow conditions, optimization-based algorithms are built around the maximization of some performance index of the system. Road managers may be interested in implementing strategies that, as they deal with congestion on the managed road, satisfy some objectives as maximization of revenue, social welfare, freeway throughput, or a combination of them [48]. A general formulation of the unconstrained optimization problem for the definition of such strategies is given by Equation (6):

$$\pi(n) = \underset{\pi(n)}{\operatorname{argmax}} \sum_{k=n}^{n+N} g(\pi(k)), \tag{6}$$

where the toll price for tolling interval $n + 1$, $\pi(n + 1)$ is the one that maximizes the accumulation of a performance index $g(\pi(k))$, chosen according to the strategy objectives, over a certain period of time, called *rolling horizon*, corresponding to the following $N$ intervals. The complete formulation of the optimization problem is given by (6) with appropriate constraints. Manifold single-objective pricing strategies with similar formulations are presented in literature: revenue-maximizing problems are described in [28–30,46], a throughput maximizing problem is formulated in [31], and an average-flow maximizing toll is presented in [8]. In [8,30,31], penalty methods are applied, replacing the constraints to ensure free-flow with penalty functions added to the objective function.

As is intuitive, strategies obtained via the maximization of a performance indicator associated with a single objective may not satisfy other possible objectives: for example, total system travel time minimization and revenue maximization are found to conflict in [40]; multi-objective approaches in literature aim to design tolls from a compromise among multiple objectives, seeking Pareto optimum, a solution characterized by the fact that "no other feasible solutions can improve at least one objective without deteriorating another" [60]. A bi-objective optimization approach is proposed in [33], where the objective function $\gamma$ is defined as the weighted sum of the system cost $c$ (in monetary units) and time $\tau$ (in time units) (7):

$$\gamma = c + w \cdot \bar{\bar{\xi}} \cdot \tau, \tag{7}$$

where $\bar{\bar{\xi}}$ is the weighted average value of time (VOT), to convert time units in monetary units, and $w$ is a non-negative weighting parameter, called *Pareto parameter*. The obtained toll expression reduces to a minimum cost scheme for $w = 0$ and to a minimum time scheme, in case of $w \to \infty$. The authors of [60] propose a toll defined by a bi-objective program for a tollable vehicular network, seeking to minimize both total delays and emissions, formulating the problem as a mathematical programming with complementarity constraints (MPCC) and solving it with a quadratic penalty-based gradient projection algorithm. A Pareto-improving queue-eliminating dynamic toll is proposed in [34].

Dynamic toll pricing problems are generally particularly difficult because of their fundamental bi-level form, where the upper level is the one picking the optimal tolls, which are subject to route-choice constraints that constitute the lower-level dynamic traffic assignment [81]. The computational complexity of the toll design problem is known to be NP-hard [81–83], even when the formulation is not explicitly dynamic [37,84–87]. As a consequence, these problems are typically solved through heuristics or meta-heuristics [81]. In [35], a bi-level dynamic second-best toll pricing model is solved through a relaxation scheme consisting in the conversion of the bi-level formulation into a single level nonlinear programming problem, which is then solved iteratively. In [36], the genetic algorithm and the method of successive average are used to solve a two-layer network dynamic congestion pricing problem. In [37], a hybrid self-adaptive gradient projection (SAGP) and artificial bee colony (ABC) algorithm are employed to solve a bi-level dynamic congestion tolling

problem, with the SAGP solving the lower level and ABC for the upper one. In [38], bi-level cellular particle swarm optimization is applied to a dynamic optimal toll problem with equilibrium constraints under demand uncertainty. Optimal dynamic pricing problems have also been formulated as Markov Decision Process and solved through approximate dynamic programming via state space aggregation [39], value function approximation with different initializations [40], a multiagent reinforcement learning algorithm [41] and policy gradient methods with tolls determined via a feedforward neural network [42].

The bi-level nature of dynamic toll pricing problems can also be described from a game theory perspective, where drivers and road managers are identified as players. At the lower level, drivers choose their paths pursuing their travel costs minimization without considering the impact of their choices on the other network users [88]. The traffic assignment resulting from drivers' selfish choices is a *Nash equilibrium* called *user equilibrium* (UE) or *Wardrop equilibrium* [39,89,90]. User equilibrium traffic assignment may not coincide with the socially best solution, identified as *System Optimum* (SO), which is characterized by minimization of the total system travel cost [86]. In this context, road managers can introduce congestion pricing at the upper level as a cooperation mechanism to minimize total system costs [91], driving a user equilibrium pattern toward a System Optimum. A Stackelberg game occurs between the road operators who act as leaders due to their dominating role in the decision process and the users who act as followers, whereas a Nash game occurs between competing road operators [92]. Dynamic toll pricing algorithms based on Stackelberg game theory are presented in [43,45].

According to [93], dynamic marginal cost tolls can stabilize transportation networks around their social optimum traffic assignment. In [94], the system optimal traffic flow pattern is obtained through piecewise linear tolls on a subset of the network links. In [51], marginal costs-based dynamic pricing strategies are showed to eliminate all the delays and minimize the total schedule deviation cost experienced by users. A dynamic toll pricing scheme to achieve system optimum on a day-to-day basis is explored in [95].

The authors of [96], analyzing a morning commute problem, compared a conventional system optimum corresponding to the minimization of the total system travel cost taking into account user heterogeneity with a time-based system optimum, where the total travel time is minimized and found that the latter is Pareto-improving compared to the former.

In summary, in this Section, we have described the main dynamic toll pricing rules proposed in the literature, classifying them depending on whether they are based on feedback control (Section 2.1) or on optimization theory (Section 2.2). The following Section is dedicated to the simulation methods that can be employed to test the pricing rules.

## 3. Overview of Simulations

The pricing rules described in Section 2 represent possible interventions on transportation networks to improve operating achievements. Real-life drivers are informed of the toll value and decide whether to pay the toll to use the tolled facility or choose an alternative transportation solution. Data about the traffic formed by the users who accepted to pay the toll are usually collected through sensors, then elaborated and used as inputs for the definition of the toll. Nevertheless, since real-life experiments are challenging to perform in the transportation engineering field, there is a need for a simulation environment that can model drivers' choices and their effects on traffic in order to test the performance of new pricing strategies in a closed-loop framework. This section is dedicated to reviewing the main simulation frameworks presented in the most recent dynamic toll pricing literature. Traffic and Driver Behavior Models are presented in Sections 3.1 and 3.2, respectively. In view of the ever-growing ecologic concerns, recent researches frequently include emissions models to simulate vehicle-driven environmental deterioration. We dedicated Section 3.3 to the discussion of these models.

### 3.1. Traffic Simulation

The traffic model is an essential part of dynamic tolling design studies; it aims to quantify some traffic characteristics corresponding to each transport alternative, depending on the traffic volumes choosing that particular transport solution. A comprehensive survey on traffic state estimation can be found in [97]. According to the scale, traffic flow models can be classified as macroscopic, mesoscopic or microscopic, with an increasing level of detail:

- macroscopic models are usually based on the analogy of traffic with fluid dynamics, thus traffic is described by the value of a few synthetic variables (flow, density and speed);
- microscopic models focus on the single vehicles' trajectories, and mesoscopic models share both the previously mentioned families' elements.

In terms of formulation in a macroscopic model, traffic state is described by vector (8):

$$H(x,t) = \begin{bmatrix} v(x,t) \\ \rho(x,t) \\ q(x,t) \end{bmatrix}, \tag{8}$$

where $v(x,t)$ is velocity (in km/h), $\rho(x,t)$ is density (in veh/km) and $q(x,t)$ is flow (in veh/h), all function of time $t$ and location $x$ on the infrastructure. Traffic flow is modeled locally through a system of partial differential equations inspired by hydrodynamics, as the system given by (9):

$$\begin{cases} \frac{\partial q(x,t)}{\partial x} + \frac{\partial \rho(x,t)}{\partial t} = 0 \\ q(x,t) = q(\rho(x,t)) \end{cases}. \tag{9}$$

The first equation of the system is a conservation law, whereas the second equation, expressing flow rate in function of density, is called fundamental diagram.

The Lighthill-Whitham-Richards (LWR) kinematic wave model [98,99], characterized by the fundamental relation (10):

$$q(x,t) = v(x,t) \cdot \rho(x,t). \tag{10}$$

This relation is considered pioneering in macroscopic traffic flow modeling [97]. This model is used to characterize traffic flow evolution on the HOT lane in [20]. A stochastic macroscopic traffic flow model built by [47] on the classical LWR model is adopted in [46] for managed lanes. An accumulator model consisting in an aggregate, lumped-parameter model, based on the LWR model, is proposed in references [17,22]. The Cell Transmission Model (CTM) [100,101] and its derivatives, which are based on the discretization of the differential equations of the LWR model, are broadly used in traffic flow modeling for dynamic tolling design. In [40,48], traffic dynamics on managed lanes is modeled through CTM. Lou [8] adopts a multi-lane hybrid CTM proposed by [32] to describe traffic dynamics in HOT facilities.

The Macroscopic fundamental diagram, also called Network Fundamental Diagram, introduced by [102] to model networks traffic dynamics on an aggregate level by considering relations between space-mean traffic flow and density in urban regions, is adopted in [18,19,36,50,51].

In microscopic models, on the contrary, traffic is described at a disaggregated level by single-vehicle trajectories as given by (11):

$$x_i = x_i(t). \tag{11}$$

In (11), $x_i(t)$ identifies the position of vehicle $i$ at time $t$. Microscopic modeling is at the base of simulation platforms that have been often used in dynamic toll pricing studies at the network level: car-following-based VISSIM [11,14–16,23,36], Paramics [21,27], CORSIM [29,52] and MITSIM [30]. A mesoscopic modeling-based simulator, DynusT,

is employed in [53] and a regional mesoscopic dynamic traffic assignment simulation environment is adopted in [54,56]. DynaMIT, a mesoscopic dynamic traffic simulator is applied in [30] toghether with microscopic simulation. Usually, these platforms can model both traffic and driver behavior.

In some studies, where the considered situation is analyzed as a bottleneck, traffic modeling is limited to a delay operator: in [13,49] traffic dynamics are described through the point-queue concept [103], assuming that the average travel time is given by the sum of a congestion-independent cruise time and a queuing delay. In [45] delays derive from the comparison between arrivals and departures curves, and a similar approach is followed in [57]. BPR (Bureau of Public Roads) volume-delay functions are adopted in [58,59]. Queuing theory is used in [24,25]. Arnott's bottleneck model is finally employed in [43]. For more details on traffic modeling, we refer the reader to [67].

As shown in Figure 5, traffic simulation is quantitatively dominated by macroscopic models, delay operators and microscopic models, which together cover almost three fourths of the analyzed methods with similar percentages, between 20% and 30%.

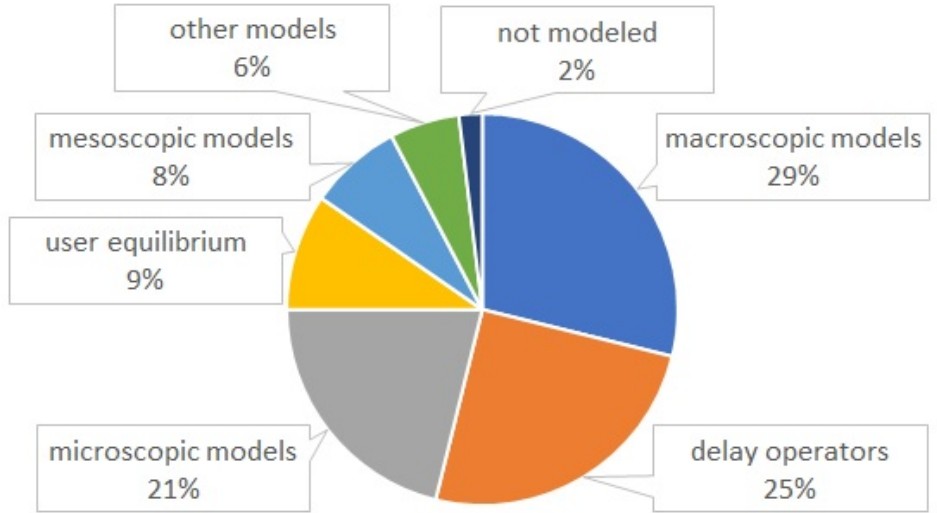

**Figure 5.** The distribution of the analyzed studies sorted by traffic simulation method.

### 3.2. Driver Behavior

A User Behavior Model reproduces people's choices among different transportation alternatives based on the alternatives' and the users' characteristics. Drivers' choices between two alternatives may be described through binary logit models, where the utility of each alternative is a function of its current travel time and toll [6,8,11,13,16,17,20,22,24,25,28,31,46,48,49]. In such cases, general utility function $U_j(t)$ for an alternative $j$ is formulated as in (12):

$$U_j(t) = \delta \cdot \pi_j(t) + \epsilon \cdot \tau_j(t) + \zeta, \tag{12}$$

where $\delta$ and $\epsilon$ represent the weighting of travel time and toll price, respectively, and $\zeta$ is an offset term used to take into account other unobservables. $\pi_j(t)$ and $\tau_j(t)$ are respectively toll price and travel time of alternative $j$ at time $t$. Other factors may be taken into account besides toll and travel time, for instance in [14], the utility is also a function of travel time reliability, which, in some cases, has been even found to be more valued than travel time savings [104]. Once defined a utility function, the probability of choosing alternative $j$ in a binary logit model is expressed by (13):

$$p_j(t) = \frac{1}{1 + e^{\Delta U(t)}}, \tag{13}$$

where $\Delta U(t)$ represents the difference in the utility function values of the two alternatives at time $t$.

Binary choices between a tolled and an untolled alternative may also be modeled through the cumulative distribution function of drivers' VOT $F(\xi)$. The users whose VOT exceeds a particular limit, called the critical VOT $\xi_{cr}(t)$, are assumed to choose the tolled alternative. The critical VOT is defined as the ratio of the toll price $\pi(t)$ and the difference in travel times $\Delta\tau(t)$ between the two alternatives at time $t$, as in (14):

$$\xi_{cr}(t) = \frac{\pi(t)}{\Delta\tau(t)}. \tag{14}$$

Indicating with $p(t)$ the proportion of drivers choosing the tolled alternative we can formulate the model as in (15):

$$p(t) = p(\xi_{cr}(t)) = 1 - F(x_{cr}(t)). \tag{15}$$

Different distribution functions are considered in the literature: a log-normal distribution is proposed in [45]; Weibull distribution is chosen for fitting in [29]; the authors of [12] adopt a simplified variant of the Burr distribution; the authors of [40] employ a discrete distribution of users' VOT; Gutman [23] assumes a Gaussian drivers' VOT probability distribution. We note that multinomial logit models may also be considered for modeling choices with more than two alternatives, whereas the driver behavior model described by (14) and (15) cannot be used in such cases. This last model, in turn, can consider drivers' heterogeneity in VOT.

Drivers choices can also be modeled through user equilibrium considerations. As mentioned in Section 2.2, the user equilibrium is a Nash equilibrium derived from drivers' selfish choices that operate to pursue travel costs minimization. In [57] the user equilibrium condition between tolled and untolled alternative for users all characterized by the same VOT is formulated as given by (16):

$$\tau_{untolled}(t) = \tau_{tolled}(t) + \pi_{tolled}(t), \tag{16}$$

where $\tau_{tolled}(t)$ and $\tau_{untolled}(t)$ indicate the travel time on the tolled alternative and on the untolled alternative at time $t$, respectively, and $\pi_{tolled}(t)$ is the toll price of the tolled alternative at time $t$. Condition (16) holds when both alternatives are used; otherwise, only the less expensive alternative is used.

Modeling the case of large networks is typically more complex than for two-alternative cases: a route-choice model usually models driver behavior in association with a departure-time-choice model. The authors of [60] present a dynamic user equilibrium, where travelers engage in a Nash-like game choosing routes and departure times aiming to minimize selfishly their travel costs, which account for travel time, early/late arrival penalties and tolls. The authors of [54,56] adopt an econometric model considering drivers' socio-economic characteristics and the network level-of-service to capture the drivers' departure time choice in consequence of the toll. In [30], a path-size logit model is used for route choice, considering the similarities between overlapping paths.

While most models assume homogeneous drivers, agent-based models may be used to take into account drivers' heterogeneity, as in [14,15,18,21,50].

In Figure 6 we may observe that logit models are quantitatively prevailing in driver behavior simulation, with user equilibrium, VOT distribution and agent-based models all present in over 10% of the studies and choice models embedded in micro-simulation traffic models (label: embedded in micro) in about 4% of the studies.

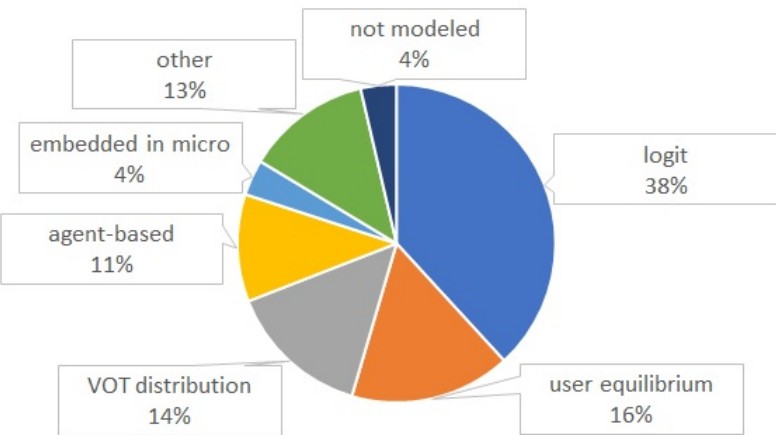

**Figure 6.** The distribution of the analyzed studies sorted by driver behavior model.

### 3.3. Externalities Quantification

Drivers are responsible for additional costs imposed upon other users and society, such as congestion, accidents, pollutants emissions and noise. These costs, called externalities, can be internalized through road charging [4]. As referred in Section 2.2, social welfare improvement, and maximization are among the objectives of dynamic pricing schemes, thus, together with the quantification of travel times, which is usually done through traffic assignment and reflects congestion, some studies include models for the quantification of other externalities.

Vehicular pollutants emissions include carbon dioxide ($CO_2$), nitrous oxides ($NO_x$), volatile organic compounds (VOC), carbon monoxide (CO), and particulate matter ($PM_{10}$) [105,106]. The emission may be estimated through emission factors (in gram/km), as in [107], or modeled in function of traffic conditions. Emission models are classified by [60] into three categories: microscopic, macroscopic and mesoscopic. In microscopic approach, the emission rates $e_i(t)$ of single vehicles $i$ (in gram/m/s), are modeled as a function of the vehicular velocity $v_i(t)$ (in km/h) and, possibly, acceleration $a_i(t)$ (in m/s$^2$), as in (17):

$$e_i(t) = f(v_i(t), a_i(t)), \tag{17}$$

while in macroscopic emission models, the average emission rate $\bar{e}(t)$ (in gram/m/s) on a road segment is expressed as a function of the average density $\bar{\rho}(t)$ in (veh/km) and average velocity $\bar{v}(t)$ (in km/h) in that segment as in (18):

$$\bar{e}(t) = g(\bar{\rho}(t), \bar{v}(t)). \tag{18}$$

Finally, mesoscopic models combine elements of the two previous approaches.

In [86], carbon dioxide emissions are quantified through a microscopic model as a velocity function only. In [108], emission rate is calculated at a macro level as a function of average segment velocity and vehicle type.

A noise emissions model is presented in [109]: noise emission levels are calculated on a macro level for each road segment due to traffic volume, heavy-duty vehicles share and maximum speeds. A grid of receivers is considered for calculating noise immission levels, which allow the quantification of the noise damages to be internalized.

The authors of [110] recently observed that when more than one externality is considered for marginal cost pricing, there exist correlation between the externalities, and it should be taken into account with proper correction factors.

Not only are externalities considered for dynamic toll schemes design, but they are also part of evaluation criteria, as in the Benefit-cost analysis of variable pricing projects in [105,106].

In summary, in this section we have presented the main components of the traffic simulation frameworks adopted in literature, grouping them into traffic models (Section 3.1), driver behavior models (Section 3.2) and externalities quantification models (Section 3.3). We now briefly discuss the most recent technology applications in the dynamic toll pricing field.

## 4. Interactions with Recent Technology Applications

The development of new communication and transportation technology is having various positive effects on road toll pricing, making it faster, easier, cheaper and more reliable [111]. In particular, connected traffic sensors enable the collection and storage of big amounts of data, which can be processed and mined to support real-time decision making [59], including dynamic toll pricing definition. Self-learning techniques have been proposed to gradually learn the parameters of the driver behavior model by the mining of the recorded traffic data [8,13,31], enabling a more accurate representation of the users' preferences, which is fundamental for price definition. Reinforcement learning algorithms for optimization of dynamic pricing strategies are employed in [41,42] and random forest predictions are included in [62].

Connected drivers and vehicles allow the definition of user-specific strategies, where toll authorities can guide users' mode and routing choices with user-dependent advice and dynamic pricing [109,112–114].

## 5. Conclusions

In this paper, we have reviewed recent literature about dynamic toll pricing focusing in particular on the studies published since 2008. We have identified the main objectives of dynamic toll pricing in keeping free-flow conditions, minimizing travel times and reducing externalities. We have observed that control-based price definition rules focus more on satisfying the first objective, while optimization-based algorithms, which represent the prevalent method, are built upon the maximization of some performance index of the system. We have identified a common structure in dynamic toll pricing studies, formed by a price definition rule together with a simulation framework composed by a model of the driver behavior, a traffic flow model and, possibly, externalities quantification models.

The scope of dynamic toll pricing varies from single infrastructures, as in the case of managed lanes facilities, where in some cases it is already implemented, to large scale networks, for which there are no known implementations at present. Managed lanes constitute the scope of most of the dynamic toll pricing schemes reviewed in this paper, followed by networks and tolled facilities.

Implemented dynamic toll pricing schemes are indeed still very few worldwide but the evolution of information technology, which allows increasingly fast and massive exchange of information, together with the progress in computation capabilities and the advancement of AI paradigms are certainly paving the way to the ideation of more complex and efficient dynamic toll pricing schemes and to new implementations, as dynamic pricing keeps emerging as a practice in multiple domains of Intelligent Transportation Systems.

**Funding:** C.L. would like to acknowledge the support of the Fundação para a Ciência e a Tecnologia (FCT), IP - Portugal for the Ph.D. Grant PD/BD/128137/2016. A.M.A. would like to acknowledge the support of the Ford-MIT Alliance.

**Institutional Review Board Statement:** Not applicable.

**Informed Consent Statement:** Not applicable.

**Data Availability Statement:** Data available within the article. All the quantitative considerations about the popularity of different scopes and methods presented in this article are determined based on the papers mentioned in Table 3.

**Conflicts of Interest:** The authors declare no conflict of interest.

## Abbreviations

Abbreviations used specifically in Table 3 are referred in its caption. The following abbreviations are used in the rest of this manuscript:

AI      artificial intelligence
HOT    High Occupancy Toll
VOT    Value of Time
ABC    Artificial Bee Colony
veh     equivalent passenger vehicles
h        hour
km     kilometer
LWR    Lighthill-Whitham-Richards
CTM    Cell Transmission Model

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

## Short Biography of Authors

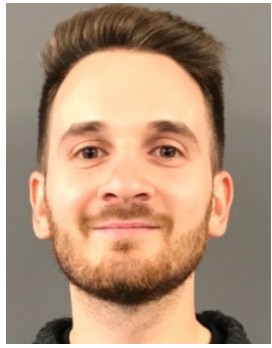

**Claudio Lombardi**, 29, received the B.S. degree in Civil Engineering from Politecnico di Milano, Milan, Italy, in 2014 and the M.S. degree in Civil Engineering from Instituto Superior Técnico and from Politecnico di Milano in 2016. He has been a visiting student in the Active-Adaptive Control Laboratory, Department of Mechanical Engineering at Massachusetts Institute of Technology during 2018–2020. He is currently pursuing the Ph.D. degree in Transportation Systems in CERIS, Instituto Superior Técnico, Lisbon, Portugal, working on dynamic toll pricing for freeways.

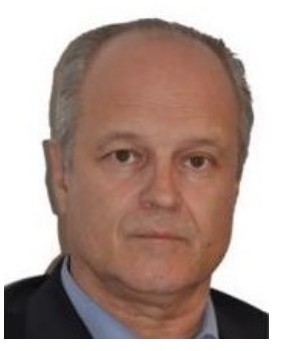

**Luís Picado-Santos**, 60, PhD, is a Full Professor of Transport and Infrastructures. He was President of the research centre CERIS (2019–2020). Luís is Director of the Doctoral Program in Transportation Systems, initiated under the MIT-Portugal joint collaboration program. He is Director of the Highways and Transport Experimental Laboratory. He is working in an international IR&D project and several short-term IR&D projects. Also supervises five PhD students. Since 1995, Luís supervised 22 concluded PhD and 63 MSc dissertations on pavement mechanics, asset management, dynamic traffic management, and road safety. In the same period, he was in charge of research and technology transfer to industry projects (14 and 20 respectively). He is the author of more than 300 international publications, including 60 articles on international peer reviewed journals (ISI and/or SCOPUS). For more than 25 years, he has had an intense consultancy activity with international and local agencies and the private sector.

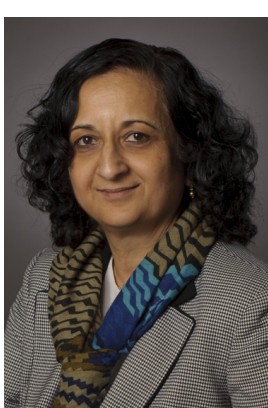

**Anuradha M. Annaswamy** is Founder and Director of the Active-Adaptive Control Laboratory in the Department of Mechanical Engineering at MIT. Her research interests span adaptive control theory and its applications to aerospace, automotive, and propulsion systems as well as cyber physical systems such as Smart Grids, Smart Cities, and Smart Infrastructures. Her research team of 15 students and post-docs is supported at present by the US Air-Force Research Laboratory, US Department of Energy, Boeing, Ford-MIT Alliance, and NSF. She has received best paper awards (Axelby; CSM), Distinguished Member and Distinguished Lecturer awards from the IEEE Control Systems Society (CSS) and a Presidential Young Investigator award from NSF. She is the author of a graduate textbook on adaptive control, co-editor of two vision documents on smart grids as well as two editions of the Impact of Control Technology report, and a member of the National Academy of Sciences Committee that published a report on the Future of Electric Power in the United States in 2021. She is a Fellow of IEEE and IFAC. She was the President of CSS in 2020.