# Peer review of "Model-Based Dynamic Toll Pricing: An Overview"

_applsci, doi:10.3390/app11114778_

Round 1

Reviewer 1 Report

This paper is well organized to the review of research results. The authors described the methods in mathematics and simulations.

I have a minor suggestion: I hope the authors can also categorize the reviewed papers in methods, models and the application of models. I see the table to show the distribution of publications by countries. It is more important to summarize and show the methods, models and the main contribution of each paper. This type of table can show a big picture of the published document.

Author Response

Dear reviewer,

Please see the attachment for the details of all the revisions.

Reviewer 2 Report

The paper presents well prepared review about dynamic toll pricing. The citations are very detailed and the scope of the papers reviewed is good and appropriate. It could be only suggested to use not only Scopus and Google but also Clarivate Analytics to find the appropriate papers. But since the number of papers reviewed is very big this isn't a problem: the review captures all main articled published in the field in recent decade. The ideas selected to review in more details are in principle correct and well reflecting the progress in the area. I could suggest to provide also quantitative characteristics about the popularity of different methods as these methods are described and analyzed in papers

Author Response

(The authors gave the same response as above.)

Reviewer 3 Report

Content
----------
The goal of this paper is to review some of the most recent research regarding design, simulation, implementation and evaluation of dynamic tolling schemes. The author identify the common elements and the differences in the approaches chosen by different authors, presenting an overview of the methods for price definition and of the simulation techniques. The author also discussed on the newest technology applications in the field.

Major comments
--------------

1. Abstract is too simple
Please add detail discussion about this field.

2. Style of reference
As a review article, my suggestion is to switch to APA style.
https://www.mdpi.com/authors/references
https://libguides.brown.edu/c.php?g=293899&p=6200917

Evaluation
--------------
Given the above, I'm in a position to major revision. 

Author Response

(The authors gave the same response as above.)

Round 2

Reviewer 3 Report

The revised version meet the standard of review paper.